# Clinical Impact of Supraclavicular Lymph Node Involvement of Stage IIIC Non-Small Cell Lung Cancer Patients

**DOI:** 10.3390/medicina57030301

**Published:** 2021-03-23

**Authors:** Sunmin Park, Won Sup Yoon, Mi Hee Jang, Chai Hong Rim

**Affiliations:** Department of Radiation Oncology, Ansan Hospital, Korea University, 123 Jeokgeum-ro, Danwon-gu, Ansan 15355, Gyeong-Gi Do, Korea; sunmini815@gmail.com (S.P.); irionyws@korea.ac.kr (W.S.Y.); 4060622@naver.com (M.H.J.)

**Keywords:** non-small cell lung carcinoma, radiochemotherapy, intensity-modulated radiotherapy, supraclavicular lymph node, NSCLC IIIC

## Abstract

*Background and Objective:* Investigations on the clinical impact of supraclavicular lymph node (SCN) involvement in stage IIIC non-small cell lung cancer (NSCLC) remain scarce. We evaluated the oncological outcomes of definitive radiochemotherapy and the clinical significance of SCN involvement. *Materials and Methods:* Between November 2009 and June 2019, a total of 40 patients with N3-positivity and NSCLC were evaluated. Most patients received concomitant chemotherapy, but six patients who received radiotherapy (RT) alone were also included. Twenty-one patients (52.5%) received 3D-conformal RT (3DCRT), and the remainder received intensity-modulated RT (IMRT). *Results:* The median follow-up duration was 10.7 months (range: 1.7–120.6 months). Median overall survival (OS) and cause-specific survival (CSS) times were 10.8 months and 16.3 months, respectively. Among the 40 patients, 17 (42.5%) had SCN involvement. SCN involvement negatively affected progression-free survival (hazard ratio (HR): 2.08, 95% confidence interval (CI): 1.04–4.17, *p* = 0.039) and local control (HR: 3.05, 95% CI: 1.09–8.50, *p* = 0.034). However, IMRT use was correlated with higher local control (HR: 0.28, 95% CI: 0.09–0.86, *p* = 0.027). Grade ≥3 esophagitis and pneumonitis accounted for 7.5% and 15.0% of all cases, respectively. A higher RT dose (mean dose: 66.6 vs. 61.7 Gy) was significantly correlated with grade ≥3 pneumonitis (*p* = 0.001). RT modality was a significant factor (*p* = 0.042, five of six cases occurred in the IMRT group). Conclusions: SCN involvement could negatively affect oncologic outcomes of stage IIIC NSCLC patients. High-dose irradiation with IMRT could increase local control but may cause lung toxicities.

## 1. Introduction

Stage III non-small cell lung cancer (NSCLC) is challenging to cure, despite the latest advances in technology. NSCLC continues to demonstrate a low rate of long-term survival. Because the oncological benefit is insignificant with surgical management, radiochemotherapy is considered the current standard modality, especially for patients with stage IIIB or higher [1,2]. In staging up to the 7th edition of the American Joint Committee on Cancer (AJCC), patients with N3 nodal involvement or T4N2 were classified as stage IIIB [3]. N3 nodal involvement has been reported as a possible factor that lowers survival in stage IIIB NSCLC patients [4,5]. In the recent staging of the 8th edition of AJCC, stage IIIB in the previous system was further subdivided into stage IIIC for patients with N3 disease. According to a recent study conducted by the International Association for the Study of Lung Cancer (IASLC) staging project, the median and 5-year rates of survival of patients with N3 disease (10 months and 9%) were significantly lower than those of N2 (17 months and 23%) [6].

A meta-analysis conducted by Auperin et al. revealed that patients with locally advanced NSCLC gained survival benefit from radiochemotherapy, which is thought to be due to the significant decrease in locoregional recurrence [7]. Currently, a dose of at least 60 Gy of radiotherapy (RT) for patients with stage III or higher NSCLC is regarded as the standard [8]. However, from the perspective of a radiation oncologist, it might not be easy to safely deliver more than 60 Gy of RT to patients with stage III or higher. In particular, patients with N3 disease inevitably have a large target volume; the presence of contralateral mediastinal or supraclavicular lymph nodes (SCNs) makes it difficult to maintain the safety dose limit to the lung or mediastinal organs (e.g., esophagus) during treatment [9,10].

Most of the previous studies have reported heterogeneous patients at stage III or higher. Because stage IIIC is a new categorization, studies regarding the clinical outcomes are scarce. Furthermore, a few studies have investigated the clinical impact of SCN involvement compared to contralateral lymph node metastases [11,12]. In this study, we report the clinical experience of N3-IIIC patients who underwent definitive radiochemotherapy in mid-sized tertiary hospitals to discuss its efficacy and feasibility, as well as the clinical impact of SCN involvement.

## 2. Materials and Methods

### 2.1. Patient Recruitment and Evaluation

A chart review was performed for NSCLC patients who underwent definitive RT between November 2009 and June 2019 (IRB number: 2020AS0244). The inclusion criteria were as follows: (1) pathologically confirmed NSCLC; (2) categorized as stage IIIC NSCLC according to the AJCC 8th staging system; (3) receipt of definitive RT with a prescribed dose of ≥60 Gy; and (4) no evidence of uncontrolled lesions at any site other than the lung at the start of RT. The vast majority of chemotherapy regimen was platinum and taxane based. The primary endpoints of the present study were overall survival (OS) and cause-specific survival (CSS). Progression-free survival (PFS), local control (LC), and grade ≥3 complications were the secondary endpoints. Complications related to RT were evaluated according to the Common Terminology Criteria for Adverse Events (CTCAE; version 4.03). Local recurrence was defined as the development of a new lesion or an increase in tumor size in the treated area (in-field failure). Disease progression was defined as the development of recurrence after RT. Distant metastases were defined as intrapulmonary out-of-field recurrence and/or distant recurrence at any site outside of the lung. Overall, PFS and recurrence-free survival were estimated from the date of RT initiation to the date of death, the last follow-up examination, or the date of tumor progression and recurrence, whichever occurred first. An equivalent dose of 2 Gy per fraction RT (EQD2) was calculated from the prescribed dose using an α/β ratio of 10. All patients were followed up with at 1 month after radiation and at 3- or 6-month intervals thereafter. At each follow-up, a detailed questionnaire on clinical status and a physical examination was administered to the patients, along with basic laboratory studies, liver and renal function tests, chest computed tomography (CT), and/or whole-body positron emission tomography/CT (PET/CT) as needed.

### 2.2. Statistical Analysis

For survival and univariate analyses, Kaplan–Meier estimation [13] and log-rank tests were performed. The Cox proportional hazards model [14] was used for multivariate analyses, using the backward elimination method. Considering the small number of patients and the possible suppression factor effect [15] (i.e., in a small series, factors that have possible mutual influence are likely to suppress the effects of other factors), all covariates evaluated in univariate analysis were assessed in multivariate analysis. Survival analyses were performed using SPSS v20.0 (IBM Corporation, Armonk, NY, USA), and multivariate analysis was performed using web-based Analysis with R 4.0 (available at https://cardiomoon.shinyapps.io/webr/, assessed from 15 December 2020 to 14 January 2021). There were no considerable missing clinical data regarding statistical analyses.

### 2.3. Radiotherapy Procedure

In our center, most patients with stage III or higher NSCLC were prescribed with either 60–66 Gy in 30 fractions or 63 Gy in 35 fractions. CT simulations were performed using the Philips Brilliance Big Bore CT scanner (Philips Healthcare, Cleveland, OH, USA). Patients were immobilized using Vac-Lok (KIKWANG MEDICAL, Seoul, Korea) and wing board (CIVCO, Kalona, IA, USA). A contrast-enhanced four-dimensional CT (4DCT) scan with 2–3 mm slice thickness was performed using the Varian RPM system (Varian Medical Systems, Palo Alto, CA) while monitoring respiratory signals. The 4DCT images were sorted into 10 phases such that the 0% respiratory phase corresponded to peak inhalation and the 50%–60% phase corresponded to peak exhalation. The gross target volume (GTV) can include the entire primary lung mass, with lymph nodes that have regional involvement contoured in the CT scan in the 50% or 60% phase, and the internal target volume (ITV) was set to be expanded by reflecting the movement in all phases. Diagnostic PET/CT was co-registered with simulation CT for target contouring if possible. Planning target volume (PTV) was set with a margin of 5–7 mm from ITV. Gating was not performed because the ITV reflected the movements of all phases in free breathing. The setup was performed by applying a tattoo to the patient, and daily pre-treatment cone-beam computed tomography (CBCT) matching was verified by a physician. Image guidance was performed by matching the bony landmark and the GTV contour.

## 3. Results

### 3.1. Patient Characteristics

Between November 2009 and June 2019, a total of 40 patients met the enrollment criteria and were included in the analysis. Most patients were male (80.0%); their median age was 67.5 years (range, 44.0–89.0 years). Radiochemotherapy was performed in the vast majority of patients (85%); six patients who were initially recommended to undergo radiochemotherapy, but received RT alone for medical reasons (not related to cancer) or patients’ willingness were also included. Most patients (87.5%) had an Eastern Cooperative Oncology Group (ECOG) performance status score of 0 or 1. Seventeen of 40 patients (42.5%) had SCN involvement, whereas the remaining patients had contralateral mediastinal involvement. A total of 52.5% of patients received 3DCRT, and the rest received IMRT. In EQD2, the mean doses (±standard deviation) delivered with 3DCRT and IMRT were 60.6 (±6.0) and 64.2 (±7.9) Gy, respectively. Patient characteristics are listed in Table 1.

### 3.2. Survival and Local Control Rates

The median follow-up duration was 10.7 months (range, 1.7–120.6 months), and 8 of the 40 patients survived during the follow-up period. Median OS and CSS times were 10.8 months (95% CI: 2.7–18.9) and 16.3 months (95% CI: 9.0–23.6), respectively (Figure 1). Median PFS was 6.5 months (95% CI: 4.5–8.5), and the LC rate was 53.5% at 1 year. Univariate and multivariate analyses revealed that there were no significant factors affecting OS. ECOG performance status and SCN involvement (*p* = 0.077 and 0.099, respectively, in multivariate analysis) appeared as marginal risk factors for CSS. Age and SCN involvement were factors influencing PFS (*p* = 0.028 and 0.039, respectively, in multivariate analysis). For LC rates, SCN involvement and RT modality were observed as significant factors (*p* = 0.034 and 0.027, respectively, in multivariate analysis). The results of the univariate and multivariate analyses are shown in Table 2 and Table 3, respectively. Clinical characteristics between patients with and without SCN involvement were not significantly different (Appendix A).

### 3.3. Treatment-Related Toxicities

Two patients (5%) in the IMRT group did not complete the planned RT (< 20 fractions) because of esophageal toxicities. Grade ≥3 pneumonitis and esophagitis were found in 6 (15.0%) and 3 (7.5%) patients, respectively (Table 4). No grade ≥3 hemoptysis was observed. A higher RT dose (mean dose: 66.6 vs. 61.7 Gy) was correlated with grade ≥3 pneumonitis (*p* = 0.001). RT modality was also a significant factor (*p* = 0.042); five of six pneumonitis cases occurred in the IMRT group (Table 5). In contrast, grade ≥3 esophagitis was not affected by EQD2 or RT modality (*p* = 0.438 and *p* = 0.928, respectively) (Appendix A). Grade 5 complications were not observed.

### 3.4. Patterns of Failure

During follow-up, 23 patients (57.5%) had treatment failure. Local failure (in-field failure) occurred in 18 (45.0%) of the 40 patients. Distant out-of-field failure was observed in 12 (30.0%) patients, and both local and distant metastases occurred in seven (17.5%) patients.

## 4. Discussion

Regarding OS, median survival time was 10.8 months in our cohort, which is comparable to the results of N3 patients mentioned in the IASLC database (10 months), wherein patients were diagnosed during 1999–2010 [6]. Although our cohort was recruited from 2010 to 2019, a similar oncologic outcome could be expected because the dose or modality of RT and concomitant chemotherapy have not changed significantly. Local failure occurred in 45% of patients, which is comparable to the result reported in other clinical series investigating patients with stage IIIB or higher [12,16], necessitating further optimization of locoregional treatment. Grade ≥3 complication rates of pneumonitis and esophagitis accounted for 15% and 7.5% of all cases, respectively, although not negligible, suggesting the feasibility of radiochemotherapy for N3-IIIC patients.

Auperin et al. [7] conducted a meta-analysis regarding concomitant radiochemotherapy for locally advanced NSCLC and reported that survival gain (hazard ratio (HR) 0.84, *p* = 0.004) was attributed to decreased locoregional recurrence (HR 0.77, *p* = 0.01) rather than distant recurrence (HR 1.04, *p* = 0.69). Therefore, dose escalation to achieve local control has been investigated in various studies, although the benefit of escalation for more than 60 Gy has yet to be clarified [17]. In our study, the use of IMRT was a significant factor that increased the local control rate (*p* = 0.027). In a descriptive analysis, 61.9% of patients who underwent 3DCRT experienced locoregional recurrence during follow-up, whereas the corresponding rate was 26.3% in those who underwent IMRT. Although dose difference was not significant between modalities (*p* = 0.105), excluding two patients who underwent incomplete RT in IMRT arms (< 20 fractions), the IMRT arm received a higher dose than the 3DCRT arm (mean 66.4 vs. 60.6 Gy, *p* = 0.002). Therefore, we assumed that dose escalation achieved with the help of more conformal modality, which is IMRT herein, might help in achieving higher local control. Notably, the IMRT arm showed higher local control, despite having a larger clinical target volume (i.e., the target volume of which is planned to receive the prescribed dose with borderline significance, mean 439 vs. 294 cc, *p* = 0.06) (Appendix A). However, the higher rate of local control leads to respiratory complication; the patients who experienced grade ≥3 pneumonitis were prescribed significantly higher dose of radiation (mean 66.6 vs. 61.7 Gy, *p* = 0.001). 

The prognostic impact of SCN involvement, compared to other mediastinal N3 diseases, is not yet well investigated. Many studies have investigated the clinical impact of nodal metastases according to location, involving a heterogeneous group of patients [12]. Furthermore, in studies according to the AJCC 7th edition or earlier staging system, stage IIIB covered not only N3 cases but also T4N2 cases [11]. Oh et al. [12]. recently reported that SCN involvement did not have a negative effect on oncologic outcomes, as compared to contralateral mediastinal involvement. In their study, OS and PFS were not significantly different according to SCN involvement (*p* = 0.679 and *p* = 0.223 for OS and PFS, respectively). However, in our study, SCN involvement was a factor negatively affecting CSS with borderline significance (HR: 1.97, *p* = 0.099), and a significant factor regarding local control (HR: 3.05, *p* = 0.034). Regarding biological structure, although lymphatic drainage from the lungs is generally ipsilateral, contralateral drainage through the subcarinal lymphatic channel is not uncommon, especially for those with disease in the lower lobes. In comparison, involvement of the SCN is regarded as a terminal lymphatic station before entering the venous system, and is therefore known to be related to tumor recurrence or metastases [18,19,20]. The current paucity of clinical literature regarding the prognostic impact of SCN involvement in stage IIIC patients necessitates further clinical research.

The small number of patients and the retrospective design are limitations of the current study. Some non-significant covariates in univariate analysis became significant in multivariate analysis, probably because of the suppression effect, which can be observed in a small case series [15]. However, the present study has the merit of providing clinical data solely for N3-IIIC patients, which has been scarce in previous literature. Further large-sized or multicenter trials can help understand the clinical course of N3-IIIC patients, including the prognostic impact of SCN involvement.

To summarize, our study showed that definitive radiochemotherapy was feasible for patients with stage IIIC. The application of IMRT with a higher dose could help enhance local control, and it possibly affected the occurrence of grade ≥3 pneumonitis. Therefore, planning optimization to reduce bystander irradiation on normal organs while maintaining a sufficient dose to tumors, or the application of particle therapy for further optimization, should be further investigated. The application of immunotherapy, which includes the use of drugs such as durvalumab [21], is expected to further increase the outcome, although none of the patients in the present study received it. The clinical impact of SCN involvement in stage IIIC patients should also be investigated to provide further optimized and personalized treatments.

## 5. Conclusions

Our study showed definitive radiochemotherapy for N3-positive NSCLC patients was feasible and yielded comparable oncologic outcomes. High dose irradiation with IMRT can increase local control, but might cause grade ≥3 pneumonitis.

## Figures and Tables

**Figure 1 medicina-57-00301-f001:**
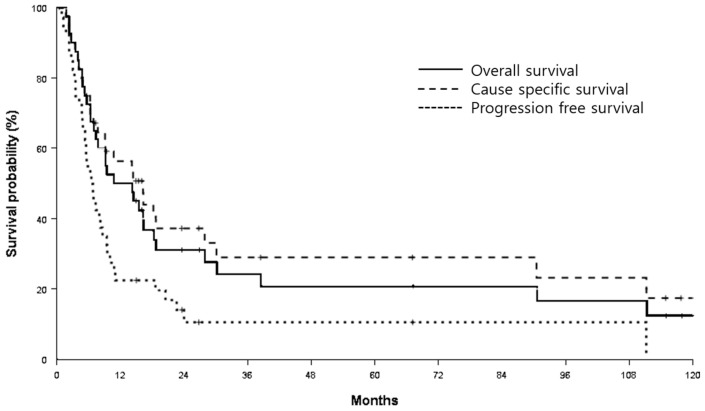
Overall survival (OS), cause-specific survival (CSS), and progression-free survival (PFS) outcomes. Median OS and CSS times were 10.8 months and 16.3 months. Median PFS time was 6.5 months.

**Table 1 medicina-57-00301-t001:** Patient characteristics.

Variables	No. of Patients (%) (Total = 40)
Sex	
Male	32 (80.0%)
Female	8 (20.0%)
Age (years)	
Median (range)	67.5 (44–89)
ECOG PS	
0 or 1	35 (87.5%)
2 or higher	5 (12.5%)
Smoking history	
Non-smoker	7 (17.5%)
Smoker (including ex-smoker)	33 (82.5%)
Current smoking	
Non-current	28 (70.0%)
Current smoker	12 (30.0%)
Histology	
Squamous cell carcinoma	26 (65%)
Adenocarcinoma	8 (20.0%)
Not specified	6 (15%)
T stage	
T1	3 (7.5%)
T2	14 (35.0%)
T3	8 (20.0%)
T4	15 (37.5%)
Nodal status	
Supraclavicular lymph node	17 (42.5%)
Contralateral lymph node	23 (57.5%)
RT modality	
3DCRT	21 (52.5%)
IMRT	19 (47.5%)
RT dose in EQD2	
3DCRT	60.6 (±6.0) Gy
IMRT	64.2 (±7.9) Gy,
Clinical target volume (cm^3^)	
Median (range)	292.0 (77.0–1015.0)

ECOG PS, Eastern Cooperative Oncology Group Performance Status; 3DCRT, 3-dimensional conformal radiotherapy; IMRT, intensity-modulated radiotherapy; EQD2, equivalent dose of 2 Gy per fraction; BED, biologically effective dose; RT, radiation therapy; NSCLC, non-small cell lung cancer.

**Table 2 medicina-57-00301-t002:** Univariate analysis of OS, CSS, PFS, and LC rates in all patients.

	*n*	Median OS (95% CI)	*p* Value	Median CSS (95% CI)	*p* Value	Median PFS (95% CI)	*p* Value	1-Year LC% (±SE)	*p* Value
ECOG PS									
0 or 1	35	14.4 (6.5–22.3)		16.4 (10.7–22.1)		6.8 (4.8–8.8)		56.8% (±9.7)	
2 or higher	5	9.1 (0–19.8)	0.213	9.1 (0–19.8)	0.117	4.8 (0.7–3.5)	0.274	0%	0.813
Age, years									
<60	9	27.9 (14.1–41.8)		27.9 (14.1–41.8)		11.0 (2.2–19.8)		55.6% (±16.6)	
≥60	31	9.1 (6.4–11.9)	0.107	10.8 (2.9–18.6)	0.277	6.3 (4.2–8.4)	0.043	53.4% (±11.5)	0.635
Sex									
Male	32	9.4 (2.4–14.5)		14.4 (3.2–25.6)		7.3 (4.7–9.9)		55.2% (±11.1)	
Female	8	15.5 (6.7–24.2)	0.776	16.3 (2.4–30.2)	0.801	5.3 (2.3–8.3)	0.151	45.0% (±18.8)	0.376
Smoking status									
Non-smoker	7	16.3 (6.9–25.7)		16.3 (1.7–30.9)		5.5 (5.0–6.0)		51.4% (±20.4)	
Smoker	33	9.4 (2.2–16.7)	0.997	14.2 (2.9–25.5)	0.94	7.3 (4.4–10.2)	0.339	53.5% (±10.9)	0.809
Histology									
Squamous	26	9.1 (4.3–13.9)		10.8 (0–23.8)		5.6 (3.2–8.0)		63.6% (±11.7)	
Others	14	14.4 (12.3–16.6)	0.91	18.7 (9.1–28.4)	0.959	6.9 (5.4–8.4)	0.809	38.6% (±14.8)	0.399
T stage									
1 or 2	17	14.2 (2.7–25.7)		16.3 (12.3–2.3)		7.3 (3.3–11.3)		70.7% (±12.6)	
3 or 4	23	9.4 (1.1–17.7)	0.937	14.4 (2.6–26.3)	0.715	6.3 (4.4–8.2)	0.359	39.8% (±12.8)	0.672
SCN involvement									
No	23	16.4 (10.0–22.7)		18.7 (3.2–34.3)		8.6 (6.4–10.8)		58.2% (±12.4)	
Yes	17	7.8 (4.0–11.5)	0.236	7.8 (4.0–11.5)	0.146	4.9 (2.5–7.3)	0.084	47.0% (±14.4)	0.189
RT modality									
3DCRT	21	14.2 (1.3–27.1)		18.3 (10.2–26.4)		8.0 (5.9–10.1)		45.9% (±12.8)	
IMRT	19	10.8 (3.2–18.3)	0.617	14.4 (6.4–22.4)	0.604	5.6 (4.5–6.7)	0.8	62.6% (±13.6)	0.1

OS, overall survival; CSS, cause-specific survival; PFS, progression-free survival; LC, local control; ECOG PS, Eastern Cooperative Oncology Group Performance Status; 3DCRT, 3-dimensional conformal radiotherapy; IMRT, intensity-modulated radiotherapy; ICI, immune checkpoint inhibitor; CI, confidence interval; RT, radiotherapy; SE, standard error; SCN, supraclavicular node.

**Table 3 medicina-57-00301-t003:** Multivariate analysis of OS, CSS, PFS, and LC rates.

	HR (95% CI)	*p* Value
*OS*		
Age	2.06 (0.84–5.03)	0.114
*CSS*		
ECOG PS	2.47 (0.91–6.74)	0.077
SCN involvement	1.97 (0.88–4.40)	0.099
*PFS*		
Age	2.79 (1.12–6.96)	0.028
SCN involvement	2.08 (1.04–4.17)	0.039
*Local control rates*		
SCN involvement	3.05 (1.09–8.50)	0.034
IMRT use	0.28 (0.09–0.86)	0.027

OS, overall survival; CSS, cause-specific survival; PFS, progression-free survival; LC, local control; ECOG PS, Eastern Cooperative Oncology Group Performance Status; SCN, supraclavicular node; IMRT, intensity-modulated radiotherapy; HR, hazard ratio; CI, confidence interval.

**Table 4 medicina-57-00301-t004:** Toxicities experienced by all patients receiving radiochemotherapy.

	Grade 2	Grade 3	Grade 4
Pneumonitis	15.0%	7.5%	7.5%
Esophagitis	50.0%	7.5%	0.0%
Hemoptysis	7.5%	0.0%	0.0%

**Table 5 medicina-57-00301-t005:** Factors related to Grade ≥3 pneumonitis.

	No Complication (*n* = 34)	Grade ≥3 Pneumonitis (*n* = 6)	*p* Value *
EQD2 (continuous)	61.7 ± 7.4	66.6 ± 0.9	0.001
EQD2			0.056
<63 Gy	19 (55.9%)	1 (16.6%)	
≥63 Gy	15 (44.1%)	5 (83.3%)	
Radiation therapy modality			0.042
3DCRT	20 (58.8%)	1 (16.7%)	
IMRT	14 (41.2%)	5 (83.3%)	
Volume of CTV	349.2 ± 226.1	531.6 ± 337.0	0.2
Mean lung dose	1436.7 ± 527.7	1439.1 ± 618.6	0.994
V20	28.4 ± 15.0	17.2 ± 2.9	0.208

EQD2, equivalent dose of 2 Gy per fraction; 3DCRT, 3-dimensional conformal radiotherapy; IMRT, intensity-modulated radiotherapy; CTV, clinical target volume; V20, volume of the normal lung receiving a dose of 20 Gy. * Chi-square test.

## Data Availability

The data that support the findings of this study are available from the corresponding author, C.H.R., upon reasonable request.

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
