# Peer review of "Clinical Impact of Supraclavicular Lymph Node Involvement of Stage IIIC Non-Small Cell Lung Cancer Patients"

_medicina, 2021, doi:10.3390/medicina57030301_

Round 1

Reviewer 1 Report

This paper reviewed the patients with stage IIIC NSCLC and suggested the unfavorable clinical impact of supraclavicular LN metastasis (SCN) on cause-specific survival and local control. 

   The authors present their institutional outcomes properly, supporting their conclusion despite this study's limitations. However, the small size of this study limits the clinical relevance of their findings. As the authors commented in the discussion, Oh et al. already reported the insignificant effect of SCN with a large study size compared to this study.  But this paper concluded that SCN is a significant factor affecting CSS and LC rate even analyzed small patients numbers. If the authors would like to insist on the clinical impact of SCN, they should present the concrete supporting data compared to that of Oh's study. Otherwise, the authors must tone down their conclusions. The paper would be more clear if it compares clinical characteristics between patients with SCN and non-SCN. 

Author Response

Reviewer 1

This paper reviewed the patients with stage IIIC NSCLC and suggested the unfavorable clinical impact of supraclavicular LN metastasis (SCN) on cause-specific survival and local control. 

The authors present their institutional outcomes properly, supporting their conclusion despite this study's limitations. However, the small size of this study limits the clinical relevance of their findings. As the authors commented in the discussion, Oh et al. already reported the insignificant effect of SCN with a large study size compared to this study.  But this paper concluded that SCN is a significant factor affecting CSS and LC rate even analyzed small patients numbers. If the authors would like to insist on the clinical impact of SCN, they should present the concrete supporting data compared to that of Oh's study. Otherwise, the authors must tone down their conclusions. The paper would be more clear if it compares clinical characteristics between patients with SCN and non-SCN. 

We fully agree with the reviewer’s considerate opinion. In the same vein with the reviewer’s opinion, we did not directly mention the clinical impact of SCN involvement in the conclusion section of the original manuscript, and we gently described in the last paragraph of the discussion (line 242-244: The clinical impact of SCN involvement in stage IIIC patients should also be investigated to provide further optimized and personalized treatments.) However, we noticed that the first sentence of abstract conclusion is bit too direct. Hence we toned down to “SCN involvement could negatively affect oncologic outcomes of stage IIIC NSCLC patients.” (from ‘SCN involvement is a significant factor that affects CSS and local control’ in the original abstract). We also performed an additional analysis on comparison of clinical characteristics between patients with and without SCN involvement. There was no significant difference in major clinical indicators, and the results is shown in Supplement Table 1 (relevant explanation in Line 144-146).

Reviewer 2 Report

The topic is interesting, and the article is quite well presented.

As the authors asserted, the strength of the research is the focus on the supraclavicular involvement, which has not been widely explored yet. However, the small number of patients, and the moderate inhomogeneity of the cohort (3D vs IMRT, CRT vs RT only, SCN vs no SCN) makes it quite difficult to draw significant conclusions, especially because only 17 patients had SPC involvement. Anyway, the article presents some valuable hints for future studies.

In my opinion, some corrections to the paper are required.

Abstract

  • both SLN and SCN acronyms were unnecessarily used in the abstract and in the paper, please choose one

Introduction

  • lines 40-43: the sentence is not fluent, and should be changed
  • line 52: since you are referring to current standard, so IMRT is probably used, it should not be too difficult to spare the spinal cord 
  • line 55: references of the "few studies" are missing

Materials and Methods

  • paragraph 2.3: was contrast media administered during simulation CT? Were simulation CT-PET available? Or were diagnostic CT-PET co-registered to simulation CT for target delineation?
  • TABLE 1: "smoking history" and "histology" numbers and percentages do not mach

Results

  • line 162: IMRT is reported as a risk factor for lung toxicity, but the dose was also higher, so it can be a confounding factor
  • paragraph 3.4: the sum of treatment failures does not match with the number reported in line 173. If patients had both in-field and out-of-field, it should be clarified

Discussion

  • lines 193-200: patients treated with IMRT also received a higher dose of RT. How can you certainly attribute the LC result to the technique instead of the dose, giving the small number of patients? Please clarify

Conclusions

  • line 240: CCRT acronym never appeared in the paper, so use the extended form

Author Response

Reviewer 2

The topic is interesting, and the article is quite well presented.

As the authors asserted, the strength of the research is the focus on the supraclavicular involvement, which has not been widely explored yet. However, the small number of patients, and the moderate inhomogeneity of the cohort (3D vs IMRT, CRT vs RT only, SCN vs no SCN) makes it quite difficult to draw significant conclusions, especially because only 17 patients had SPC involvement. Anyway, the article presents some valuable hints for future studies.

In my opinion, some corrections to the paper are required.

Abstract

  • both SLN and SCN acronyms were unnecessarily used in the abstract and in the paper, please choose one
  • We thank you for your insightful comment. As the reviewer pointed out, the entire manuscript was modified by unifying it with SCN. Thank you for pointing out.

Introduction

  • lines 40-43: the sentence is not fluent, and should be changed
  • Agreeing your comment, we adjusted the sentence to convey clear meaning and context.
  • line 52: since you are referring to current standard, so IMRT is probably used, it should not be too difficult to spare the spinal cord 
  • We appreciate and agree your keen point. As such, we adjusted ‘spinal cord’ to ‘mediastinal organs (E.g. esophagus)’, considering that the present study investigated pneumonitis, esophagitis, and hemoptysis as possible main complications.
  • line 55: references of the "few studies" are missing
  • We appreciate your keen point and added the relevant references.

Materials and Methods

  • Paragraph 2.3: was contrast media administered during simulation CT? Were simulation CT-PET available? Or were diagnostic CT-PET co-registered to simulation CT for target delineation?
  • We appreciate for the insightful comments. We performed contrast enhanced 4D simulation CT and added this to the manuscript. Unfortunately, simulation PET-CT was not implemented in our institution. However, diagnostic PET/CT was co-registered with simulation CT for target contouring. We added relevant explanation at line 101 and line 108-109.
  • TABLE 1: "smoking history" and "histology" numbers and percentages do not match
  • We appreciate and agree your comments. There was a mistake in calculating the number of patients and percentage in the "smoking history" and "histology" columns. As you mentioned, all these have been corrected to reflect the correct numbers (Table 1).

Results

  • line 162: IMRT is reported as a risk factor for lung toxicity, but the dose was also higher, so it can be a confounding factor
  • We appreciate and agree your consideration. It is clear that we cannot assess affections of RT modalities and dose separately in the present study. Therefore, we suppose that this query should be discussed more in the discussion section. From line 200 to 203, we explained that IMRT arm received a higher dose than 3DCRT arm, when excluding two patients who underwent incomplete RT. In line 208-209 of original manuscript, it was written suggesting that the pneumonitis was correlated with IMRT modality (original manuscript: “5 out of 6 cases of grade ≥3 pneumonitis occurred in the IMRT arm”); we adjusted the sentence to suggest that such complications were caused by higher dose rather than modality itself (revised manuscript line 209-210: “the patients who experienced grade ≥3 pneumonitis were prescribed significantly higher dose of radiation (mean 66.6 Vs. 61.7 Gy, p=0.001).).
  • Paragraph 3.4: the sum of treatment failures does not match with the number reported in line 173. If patients had both in-field and out-of-field, it should be clarified.
  • First of all, thank you for your keen comments. In the manuscript, a total of 23 failures were found. Local (in-field) failure was found in 18, distant (out-of-field) failure was found in 12, and both were found in 7 people (diagram is as follows). I have already described both in-field and out-of-field in the text, but it doesn't seem to be noticeable. Since all the information has already been provided, no additional description has been made.

Discussion

  • lines 193-200: patients treated with IMRT also received a higher dose of RT. How can you certainly attribute the LC result to the technique instead of the dose, giving the small number of patients? Please clarify
  • We appreciate and agree your point. Although we did not intend to attribute the LC result to IMRT (we supposed LC result is due to higher dose rather than modality, in the same vein with your opinion), we agree the necessity of clearer context. Therefore, we added a sentence in line 203-205: “We assumed that dose escalation achieved with help of more conformal modality, which is IMRT herein, might help achieving higher local control “

Conclusions

  • line 240: CCRT acronym never appeared in the paper, so use the extended form
  • We thank you for your insightful point. 'CCRT' is a term that is not used in the manuscript, and we have made the mistake of using it in our conclusions. So, the 'CCRT' in the conclusion was changed to 'radiochemotherapy'.